# Arctic-SnowCoder: Demystifying High-Quality Data in Code Pretraining

## Abstract

Recent studies have been increasingly demonstrating that high-quality data is crucial for effective pretraining of language models. However, the precise definition of "high-quality" remains underexplored. Focusing on the code domain, we introduce Arctic-SnowCoder-1.3B, a data-efficient base code model pretrained on 555B tokens through three phases of progressively refined data: (1) *general pretraining* with 500B standard-quality code tokens, preprocessed through basic filtering, deduplication, and decontamination, (2) *continued pretraining* with 50B high-quality tokens, selected from phase one by a BERT-style quality annotator trained to distinguish good code from random data, using positive examples drawn from high-quality code files, along with instruction data from Magicoder and StarCoder2-Instruct, and (3) *enhanced pretraining* with 5B synthetic data created by Llama-3.1-70B using phase two data as seeds, adapting the Magicoder approach for pretraining. Despite being trained on a limited dataset, Arctic-SnowCoder achieves state-of-the-art performance on Big-CodeBench, a coding benchmark focusing on practical and challenging programming tasks, compared to similarly sized models trained on no more than 1T tokens, outperforming Phi-1.5-1.3B by 36%. Across all evaluated benchmarks, Arctic-SnowCoder-1.3B beats StarCoderBase-3B pretrained on 1T tokens. Additionally, it matches the performance of leading small base code models trained on trillions of tokens. For example, Arctic-SnowCoder-1.3B surpasses StarCoder2-3B, pretrained on over 3.3T tokens, on HumanEval+, a benchmark that evaluates function-level code generation, and remains competitive on BigCodeBench. Our evaluation presents a comprehensive analysis justifying various design choices for Arctic-SnowCoder. Most importantly, we find that the key to high-quality data is its consistency with the distribution of downstream applications.

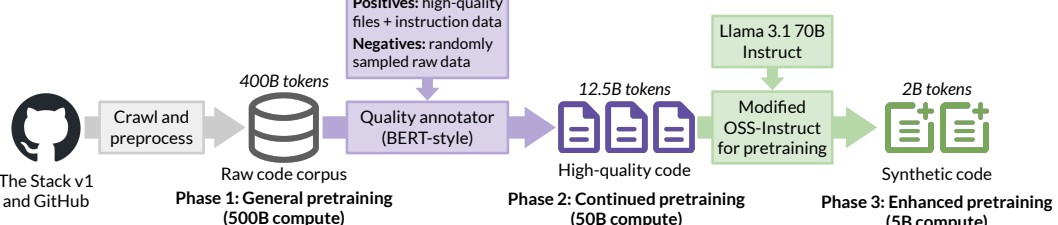

Figure 1: The three-phase pretraining of Arctic-SnowCoder-1.3B using progressively higher-quality data, sourced from the same raw code corpus.

## 1 Introduction

Pretraining large language models (LLMs) has generally relied on vast quantities of data. This emphasis on data volume is especially true in specialized domains like code, where researchers obtain massive code pretraining datasets by crawling platforms like GitHub (Li et al., 2023a; Rozière et al., 2024; Guo et al., 2024; Lozhkov et al., 2024; Mishra et al., 2024; DeepSeek-AI et al., 2024b). Recent studies, however, have increasingly showed that high-quality data is crucial for effective

pretraining (DeepSeek-AI et al., 2024a; Penedo et al., 2024; Li et al., 2024; Abdin et al., 2024), including the code domain (Gunasekar et al., 2023; Li et al., 2023b; DeepSeek-AI et al., 2024b).

In the general domain, researchers have explored various techniques to curate high-quality pretraining data for language models. FineWeb-Edu (Penedo et al., 2024) uses a linear regressor built on `Snowflake-arctic-embed-m` (Merrick et al., 2024) embeddings to assess the educational value of web pages and select high-quality content, while the DCLM (Li et al., 2024) approach employs a `fastText`-based (Bojanowski et al., 2017) filter trained on positive examples from high-quality online sources (Wei, 2024) and instruction data (Wei et al., 2024b), and random negative web pages to identify high-quality text. These model-based quality filters have been shown to significantly enhance language model performance on downstream tasks, compared to using unfiltered, large-scale datasets. Similarly, researchers have recognized the importance of high-quality code data for pretraining, with Phi-1 (Gunasekar et al., 2023) using a random forest classifier on CodeGen (Nijkamp et al., 2023) embeddings to select educational code samples, and DeepSeek-Coder-V2 (DeepSeek-AI et al., 2024a) employing a multi-stage `fastText`-based (Bojanowski et al., 2017) pipeline to recall web-related code data and high-quality code from GitHub, achieving state-of-the-art coding performance.

In this paper, we introduce Arctic-SnowCoder-1.3B, a high-performing small code model created by a novel three-step training methodology focused on progressive improvements in data quality. As a result of this methodology, Arctic-SnowCoder-1.3B outperforms StarCoderBase-3B (Li et al., 2023a) across all evaluated benchmarks and exceeds Phi-1.5-1.3B (Li et al., 2023b) by 36% on the complex and practical BigCodeBench benchmark (Zhuo et al., 2024), a benchmark that truly matters for real-world programming. As shown in Figure 1, Arctic-SnowCoder is developed through a three-stage, data-efficient pretraining process that progressively refines the quality of the data used. The first stage involves general pretraining for a 500B token horizon using 400B unique raw code data, which have been preprocessed through basic filtering, deduplication, and decontamination. The 400B raw corpus is primarily derived from the coding data used to train Snowflake Arctic (Snowflake AI Research, 2024), combining cleaned The Stack v1 (Li et al., 2023a) and GitHub crawls. This is followed by continued pretraining on 50B tokens, utilizing a smaller, high-quality subset of 12.5B code files, repeated four times. The high-quality tokens are selected from phase one by a BERT-based (Devlin et al., 2019) quality annotator trained to distinguish good code from random data, using positive examples drawn from publicly available high-quality code files (Wei, 2024), along with instruction data from Magicoder (Wei et al., 2024b) and StarCoder2-Instruct (Wei et al., 2024a). Finally, the model undergoes an enhanced pretraining phase for 5B tokens, leveraging roughly 2B synthetic data generated by Llama-3.1-70B (Dubey et al., 2024). This process uses the phase two data as seeds and adapts the OSS-Instruct methodology from Magicoder (Wei et al., 2024b) by transforming lower-quality seed code into high-quality code documents. Notably, all training phases of Arctic-SnowCoder derive data from the same raw pretraining corpus, ensuring that minimal new knowledge is introduced.

Arctic-SnowCoder-1.3B achieves state-of-the-art results on BigCodeBench (Zhuo et al., 2024), a coding benchmark focusing on practical and challenging programming tasks, among models of similar size trained with ≤ 1T tokens. Particularly, it outperforming Phi-1.5-1.3B (Li et al., 2023b) by 36%. Despite being trained on 555B tokens, compared to other state-of-the-art small code models trained on trillions of tokens, Arctic-SnowCoder matches or surpasses the performance of these models on several benchmarks. For instance, Arctic-SnowCoder-1.3B beats StarCoderBase-3B (Li et al., 2023a), trained on over 1T tokens, across all evaluated benchmarks. Arctic-SnowCoder-1.3B outperforms StarCoder2-3B (Lozhkov et al., 2024), trained on over 3T tokens, on HumanEval+ (Chen et al., 2021; Liu et al., 2023) (28.0 vs. 27.4), a benchmark evaluating function-level code generation, while remaining competitive on BigCodeBench (19.4 vs. 21.4). We conduct comprehensive ablation studies to validate the design decisions behind training Arctic-SnowCoder:

- First, our findings indicate that, in general pretraining, organizing file-level data into repositories after partitioning by programming language significantly outperforms the approach of grouping data solely by repository names.

- Additionally, we determine the optimal learning rate schedule, which involves a re-warmup phase followed by linear decay, as well as the ideal repetition of high-quality data during continued pretraining, which we find to be four times.

- More importantly, our comparisons of model-based quality annotators, trained on various data combinations, highlight that the consistency of pretraining data distribution and downstream tasks is crucial for achieving superior performance.

In summary, we make the following contributions:

- We introduce Arctic-SnowCoder-1.3B, a high-performing small code model trained on 555B tokens that benefits from progressive improvements in data quality.
- We demonstrate that high-quality data and synthetic data can significantly improve the model performance despite being seeded from the same raw corpus.
- For the first time, we demystify the notion of data quality in code pretraining by systematically comparing model-based quality annotators trained on different data combinations.
- We provide practical insights into optimal design choices for repo-level grouping in general pretraining, and optimal learning rate schedules and repetitions of high-quality data during continued pretraining, providing practical guidelines for future model development.

## 2 ARCTIC-SNOWCODER

In this section, we provide a detailed explanation of the training methodology used for Arctic-SnowCoder-1.3B, as illustrated in Figure 1. We begin by discussing the composition of the raw training data in §2.1, followed by an overview of the general pretraining phase in §2.2. Next, we describe the continued pretraining process using high-quality data in §2.3, and finally, we elaborate on the enhanced pretraining with synthetic data in §2.4. The model architecture is based on Llama-2 (Touvron et al., 2023), with specific details provided in Table 1.

Table 1: Model architecture details of Arctic-SnowCoder.

| Parameter | Arctic-SnowCoder-1.3B |
|---|---|
| hidden_dim | 2048 |
| ffn_hidden_dim | 5632 |
| num_heads | 16 |
| num_kv_heads | 16 |
| num_layers | 24 |
| vocab_size | 64000 |
| seq_len | 8192 |
| positional_encodings | RoPE (Su et al., 2023) |
| tie_embeddings_and_output_weights | True |

### 2.1 RAW DATA

The raw pretraining data used to train Arctic-SnowCoder-1.3B consists exclusively of code, primarily derived from the coding data used to train Snowflake Arctic (Snowflake AI Research, 2024). This data combines cleaned versions of The Stack v1 (Li et al., 2023a) and GitHub crawls. From this data, we select 18 popular programming languages for training, similar to StarCoder2-3B (Lozhkov et al., 2024). These languages include Python, Java, C++, C, JavaScript, PHP, C#, Go, TypeScript, SQL, Ruby, Rust, Jupyter Notebook, Scala, Kotlin, Shell, Dart, Swift, amounting to a total of 400B unique tokens.

### 2.2 GENERAL PRETRAINING

In general pretraining, the model is trained for 500B tokens with a sequence length of 8,192 and a batch size of 512 using Adam (Kingma & Ba, 2017). The learning rate follows a cosine decay after a linear warmup of 600 iterations. We set the maximum learning rate to $5.3 \times 10^{-4}$ and the minimum to $5.3 \times 10^{-5}$, following DeepSeek-Coder (Guo et al., 2024). In this phase, we use the entire 400B raw data without applying additional quality filtering. We start by partitioning code files by

programming language, grouping them by repository, and then concatenating them in random order, similar to the StarCoder2 (Lozhkov et al., 2024) approach. In §3.3, we show the advantage of first partitioning code files by programming language. We name the model produced by this phase as Arctic-SnowCoder-alpha.

## 2.3 CONTINUED PRETRAINING WITH HIGH-QUALITY DATA

After general pretraining, we continue pretraining Arctic-SnowCoder-alpha with 50B high-quality tokens sourced from the same raw pretraining corpus. The 50B high-quality tokens are formed by repeating 12.5B top-percentile code file tokens for 4 times scored by our code quality annotator. Inspired by FineWeb-Edu (Penedo et al., 2024) and DCLM (Li et al., 2024), we train a linear classi-fication head on top of `Snowflake-arctic-embed-m` (Merrick et al., 2024), a state-of-the-art embedding model based on BERT (Devlin et al., 2019). The training data comprises 300k posi-tive examples, sampled from a blend of 220k high-quality open-source code files (Wei, 2024), 80k high-quality instruction data from Magicoder (Wei et al., 2024b) and StarCoder2-Instruct (Wei et al., 2024a), and 300 randomly selected code documents from the pretraining corpus. Prior research on code quality, such as Phi-1 (Gunasekar et al., 2023), often overemphasizes the "educational value" of code, skewing models towards simpler benchmarks like HumanEval+ (Chen et al., 2021; Liu et al., 2023). In §3.2, we show that our annotation leads to a more balanced enhancement of model capabilities. Furthermore, given that these code documents typically exceed 1000 tokens, surpass-ing the BERT context window size of 512, we improve over FineWeb-Edu's pipeline to calculate the score for each file by averaging the scores from the top, middle, and bottom sections as produced by the quality annotator. In this phase, we rewarmup the learning rate for 1000 iterations from 0 to $5.3 \times 10^{-4}$, the maximum pretraining learning rate, followed by a linear decay to 0. The model pro-duced in this phase is referred to as Arctic-SnowCoder-beta. In §3.4, we perform a comprehensive analysis that validates all of our design choices.

## 2.4 ENHANCED PRETRAINING WITH SYNTHETIC DATA

In the enhanced pretraining stage, we generate even higher-quality data than in continued pretraining leveraging Llama-3.1-70B-Instruct (Dubey et al., 2024) and increase the Python mix ratio to approx-imately 50% while keeping the proportions of the other languages unchanged. Phi-1 (Gunasekar et al., 2023) demonstrates that synthetic, textbook-like pretraining data can significantly enhance model performance. However, overemphasis on such data risks skewing the model's distribution, potentially impairing its effectiveness in real-world coding tasks. For example, we show in §3.2 that Phi-1.5 excels in HumanEval+ (Chen et al., 2021; Liu et al., 2023) and MBPP+ (Austin et al., 2021; Liu et al., 2023), which resemble textbook exercises, but performs less effectively on the more com-plex and practical coding tasks in BigCodeBench (Zhuo et al., 2024). To address this, we adapt the OSS-Instruct method from Magicoder (Wei et al., 2024b) for pretraining purposes. Originally, OSS-Instruct was originally designed to generate realistic instruction-tuning data by prompting a model to create question-answer pairs inspired by open-source code snippets. In contrast, we produce high-quality synthetic pretraining data by using Llama-3.1-70B-Instruct to generate high-quality and problem-solving oriented code files, seeded with code documents scored in the top percentile during the continued pretraining phase. In §3.2, we conduct an extensive evaluation to demonstrate that each pretraining phase significantly outperforms the previous one, highlighting the effectiveness of progressively enhancing data quality.

## 3 EXPERIMENTS

In this section, we compare Arctic-SnowCoder with state-of-the-art small language models and show performance boost over each pretraining stage (§3.2), evaluate two strategies of forming repo-level data in general pretraining (§3.3), and perform detailed ablation to justify our design choices in continued pretraining (§3.4).

## 3.1 EXPERIMENTAL SETUP

We consider the following four diverse programming benchmarks to comprehensively evaluate the code generation capability of different code models:

**HumanEval+ and MBPP+ (Liu et al., 2023).** HumanEval (Chen et al., 2021) and MBPP (Austin et al., 2021) are the two most widely-used benchmarks for function-level code generation. We adopt their augmented version powered by EvalPlus (Liu et al., 2023), with 80×/35× more test cases for rigorous evaluation. HumanEval+ and MBPP+ include 164 and 378 coding problems, respectively.

**EvoEval (Xia et al., 2024)** is a program synthesis benchmark suite created by evolving existing benchmarks into different targeted domains. We employ its five default transformation categories, namely `difficult`, `creative`, `subtle`, `combine` and `tool_use`, totaling 500 tasks.

**BigCodeBench (Zhuo et al., 2024)** evaluates LLMs with practical and challenging programming tasks. It has 1140 programming tasks, where each task in BigCodeBench is created through human-LLM collaboration, where the task quality is ensured by human experts.

We incorporate HumanEval+, MBPP+, EvoEval, and BigCodeBench for baseline comparison in §3.2. For the subsequent ablation studies in §3.3 and §3.4, we include the base versions of HumanEval and MBPP while omitting BigCodeBench for faster evaluation. Throughout the experiments, we report the pass@1 metric (Chen et al., 2021) using greedy decoding.

## 3.2 BASELINE COMPARISON AND EFFECTIVENESS OF THREE-STAGE PRETRAINING

Table 2: Comparing Arctic-SnowCoder with state-of-the-art small language models ($< 3B$), divided by whether training compute $> 1T$ tokens. Arctic-SnowCoder-alpha and Arctic-SnowCoder-beta are checkpoints after general pretraining and continued pretraining with high-quality data, respectively. Arctic-SnowCoder is the final checkpoint after enhanced pretraining with synthetic data.

| Model | Training compute | HumanEval+ | MBPP+ | EvoEval | BigCodeBench |
|---|---|---|---|---|---|
| StableCode-3B (Pinnaparaju et al., 2024) | 1.3T | 26.2 | 43.9 | 18.6 | 25.9 |
| StarCoder2-3B (Lozhkov et al., 2024) | 3.3T to 4.3T | 27.4 | **49.2** | 19.0 | 21.4 |
| Granite-Code-Base-3B (Mishra et al., 2024) | 4.5T | 29.3 | 45.8 | **19.8** | 20.0 |
| CodeGemma-2B-v1.0 (Team et al., 2024) | 3T + 1T | 18.3 | 46.3 | 15.4 | 23.9 |
| CodeGemma-2B-v1.1 (Team et al., 2024) | 3T + 500B | **32.3** | 48.9 | **19.8** | **28.0** |
| Qwen1.5-1.8B[1] (Yang et al., 2024a) | 3T | 19.5 | 28.3 | 5.0 | 6.3 |
| Qwen2-1.5B[1] (Yang et al., 2024a) | 7T | 31.1 | 38.4 | 17.2 | 16.5 |
| DeepSeek-Coder-1.3B (Guo et al., 2024) | 2T | 28.7 | 48.1 | 19.2 | 22.2 |
| StarCoderBase-3B (Li et al., 2023a) | 1T | 17.7 | 36.8 | 11.6 | 5.9 |
| SmolLM-1.7B (Allal et al., 2024) | 1T | 15.9 | 34.7 | 10.0 | 2.5 |
| Phi-1.5-1.3B (Li et al., 2023b) | 150B | **31.7** | **43.7** | **20.6** | **14.3** |
| Arctic-SnowCoder-alpha-1.3B | 500B | 14.0 | 27.8 | 7.4 | 10.3 |
| Arctic-SnowCoder-beta-1.3B | 500B + 50B | 21.3 | 34.7 | 12.8 | 12.3 |
| Arctic-SnowCoder-1.3B | 550B + 5B | **28.0** | **42.9** | **18.0** | **19.4** |

[1] We remove trailing newlines from prompts in most HumanEval (+) and EvoEval evaluations. However, for Qwen1.5-1.8B and Qwen2-1.5B, we keep them due to their high sensitivity ($>15$ points drop) to newlines.

Table 2 presents a comprehensive comparison of various small language models (less than 3B parameters) across multiple coding benchmarks, categorized by whether their training compute exceeds 1T tokens. Notably, Arctic-SnowCoder demonstrates exceptional performance, particularly given its limited training data. Arctic-SnowCoder-1.3B achieves state-of-the-art performance on BigCodeBench compared to similarly sized models trained on no more than 1T token, significantly outperforming StarCoderBase-3B, SmolLM-1.7B, and Phi-1.5-1.3B. Particularly, although Phi-1.5-1.3B has an advantage in "textbook-like" benchmarks such as HumanEval+, MBPP+, and EvoEval, Arctic-SnowCoder-1.3B outperforms Phi-1.5-1.3B by 36% on the more complex and practical BigCodeBench. Also, Arctic-SnowCoder-1.3B beats StarCoderBase-3B, the predecessor of StarCoder2-3B trained on 1T tokens, across all evaluated benchmarks. Despite being trained on only 555B tokens, on HumanEval+, Arctic-SnowCoder-1.3B rivals and even surpasses models that have undergone significantly more extensive training, such as

StarCoder2-3B, StableCode-3B, CodeGemma-2B-v1.0, and Qwen1.5-1.8B. On EvoEval and Big-CodeBench, Arctic-SnowCoder remains competitive. Additionally, the table highlights the consistent improvement of Arctic-SnowCoder across its training phases: Arctic-SnowCoder-alpha, Arctic-SnowCoder-beta, and the final Arctic-SnowCoder. Each phase builds on the previous one, with Arctic-SnowCoder achieving the highest scores in all benchmarks. This steady enhancement emphasizes the crucial role of high-quality and synthetic data in the final phase. Despite starting with the same data, each iteration of Arctic-SnowCoder narrows the gap with state-of-the-art models, demonstrating the efficacy of the overall training approach.

## 3.3 REPO-LEVEL DATA IN GENERAL PRETRAINING

In the general pretraining phase, we adopt StarCoder2's approach to group file-level data randomly into repositories through a random concatenation of file contents (Lozhkov et al., 2024). In Table 3, we study two methods: (1) grouping files just by repository names, meaning that each training document can be a mix of multi-lingual code files if the repository is written in different languages, and (2) partitioning files into different programming languages before grouping them into repositories, meaning that each training document only focuses on one single language.

Table 3: Comparison of two methods for grouping repo-level data for pretraining. (1) "Group by repo" treats each repository as a single training unit with possibly mixed languages, and (2) "Group by language and repo" partitions data by programming language before grouping by repository.

| Setting | HumanEval (+) | MBPP (+) | EvoEval |
|---|---|---|---|
| Group by repo | 12.8 (10.4) | 30.7 (25.9) | 7.0 |
| Group by language and repo | **17.1 (15.9)** | **33.9 (27.8)** | **7.4** |

We can observe that the second approach, which we finally adopt in general pretraining, performs significantly better than the first one. The primary reason for enhanced performance when grouping by language before the repository is that grouping by repositories can result in training instances containing mixed file types, such as configuration files and programming files. During training, we align the compute, meaning that the "grouping by repositories" approach processes fewer tokens specifically from programming files. Additionally, since files are randomly ordered, code files from different languages are often unrelated. Consequently, each training example may include two entirely unrelated files, which can negatively affect learning.

A promising hybrid approach could involve grouping files by language within each repository. This method ensures that training examples can include multiple programming language files while maintaining the cohesion of files in the same language within each group.

## 3.4 DESIGN CHOICES IN CONTINUED PRETRAINING

In continued pretraining, we source high-quality tokens from our pretraining corpus and train an improved base model. To obtain high-quality tokens, a model-based quality annotator is employed. In this section, we experiment with various design choices of our approach, including the training data used for the annotator, the learning rate used in continued pretraining, and the optimal repetitions of high-quality tokens.

### 3.4.1 MODEL-BASED QUALITY ANNOTATOR

Similar to FineWeb-Edu (Penedo et al., 2024), we train a linear head on top of the `Snowflake-arctic-embed-m` (Merrick et al., 2024) embedding model to score each code file. In Table 4, we experiment with 4 variants:

- ANN-EDU: We prompt Mixtral-8x7B-Instruct (Jiang et al., 2024) to annotate the educational value of each code file (1 to 5). 400k annotations are used to train a linear regression head. For the following variants, similar to DCLM (Li et al., 2024), we sample negative documents randomly and change the positive parts only. We equip the embedding model with a linear classification head.

- ANN-INS: Positives are a mix of 100k educational data (3.5+) bootstrapped from ANN-EDU and 100k high-quality instruction data from Magicoder (Wei et al., 2024b) and StarCoder2-Instruct (Wei et al., 2024a).
- ANN-HQ: Positives are 220k open-source, synthetic, high-quality code files (Wei, 2024).
- ANN-HQINS: Positives are a mix of 220k ANN-HQ training data and 80k instruction data from Magicoder (Wei et al., 2024b) and StarCoder2-Instruct (Wei et al., 2024a).

Table 4: Comparison of downstream performance by applying model-based quality annotators trained with different recipes to 10B continued pretraining.

| Annotator | Training data | HumanEval (+) | MBPP (+) | EvoEval |
|---|---|---|---|---|
| Pretrained model (no continued pretraining) | | 17.1 (15.9) | 33.9 (27.8) | 7.4 |
| Continued pretraining on random 10B tokens | | 15.9 (12.8) | 30.7 (23.3) | 8.0 |
| ANN-EDU | 400k Mixtral annotations for educational scores (0–5) | 19.5 (16.5) | 27.8 (22.2) | 10.4 |
| ANN-INS | 100k high ANN-EDU + 100k instruction data from Magicoder (Wei et al., 2024b) and StarCoder2-Instruct (Wei et al., 2024a) | 21.3 (18.3) | 37.3 (29.9) | 10.4 |
| ANN-HQ | 220k open-source, synthetic high-quality code files (Wei, 2024) | 19.5 (16.5) | 33.9 (26.7) | 9.2 |
| ANN-HQINS | 220k ANN-HQ data mixed with 80k instruction data | **22.0 (18.3)** | **40.2 (33.1)** | **11.6** |

After training the annotators, we first apply each annotator to the entire pretraining corpus to obtain a score for each file. Unlike FineWeb-Edu, which only scans the top 2k characters, we scan the top, middle, and bottom parts of a code file and average the scores. We then rank the code files per language based on these scores and select the top percentile of documents until we reach approximately 10 billion tokens. We maintain the same mix ratio as used in pretraining. The table shows that ANN-HQINS, which combines both high-quality files and instruction data, achieves the best downstream performance.

To understand the underlying factor that causes the performance difference, we conduct an additional analysis in Figure 2. For each annotator, we create a validation dataset with positives from code solution benchmarks and negatives from random pretraining data not seen during training. We use the ROC-AUC (Bradley, 1997) (Area Under the Receiver Operating Characteristic Curve) score to evaluate how well the annotator ranks benchmark data. The figure illustrates the correlation between per-benchmark ROC-AUC scores and benchmark pass rates. There is an almost consistent trend: higher ROC-AUC scores lead to better benchmark performance. A good ROC-AUC score indicates that the annotator effectively shapes the distribution of downstream tasks. Thus, the key to high-quality data is essentially the alignment with downstream application distributions.

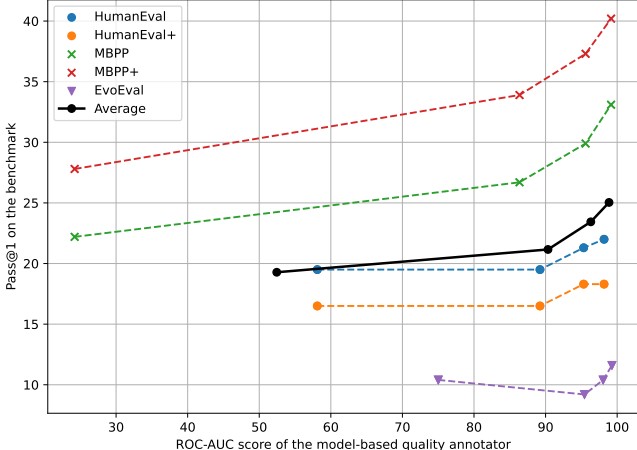

Figure 2: Correlation between annotator ROC-AUC score and benchmark pass@1.

### 3.4.2 LEARNING RATE SCHEDULE

We also study the effect of different learning rate schedules for continued pretraining in Table 5, including (1) a linear annealing starting from the minimum pretraining learning rate to zero, (2) a constant schedule using the minimum pretraining learning rate, and (3) a re-warmup to the maximum pretraining learning rate followed by a linear decay to zero. Empirically, we find that the re-

Table 5: Comparison of different learning rate schedules in 10B continued pretraining using ANN-HQINS. Here $\texttt{MIN\_LR} = 5.3 \times 10^{-5}$ and $\texttt{MAX\_LR} = 5.3 \times 10^{-4}$.

| Setting | Schedule | HumanEval (+) | MBPP (+) | EvoEval |
|---|---|---|---|---|
| Pretraining | $0 \to \texttt{MAX\_LR} \to \texttt{MIN\_LR}$ | 17.1 (15.9) | 33.9 (27.8) | 7.4 |
| Linear | $\texttt{MIN\_LR} \to 0$ | 18.3 (16.5) | 37.0 (30.4) | 9.8 |
| Constant | $\texttt{MIN\_LR} \to \texttt{MIN\_LR}$ | 20.7 (18.3) | 39.4 (31.7) | 9.4 |
| Re-warmup | $0 \to \texttt{MAX\_LR} \to 0$ | **22.0 (18.3)** | **40.2 (33.1)** | **11.6** |

warmup approach performs the best and use it consistently in all the other experiments with respect to continued pretraining.

### 3.4.3 REPETITIONS OF HIGH-QUALITY DATA

Finally, we scale up the token horizon from 10 billion to 50 billion in continued pretraining. One remaining question to address is determining the optimal repetitions for high-quality tokens. We experiment with repetitions ranging from 1 to 5, as shown in Table 6, by selecting the top percentile tokens ranked by ANN-HQINS. In this context, the top percentile tokens are the highest quality

Table 6: Downstream performance with varying repetitions of high-quality data in 50B continued pretraining using ANN-HQINS.

| Repetition pattern | HumanEval (+) | MBPP (+) | EvoEval |
|---|---|---|---|
| Pretrained | 17.1 (15.9) | 33.9 (27.8) | 7.4 |
| $1 \times 10.0B$ | 22.0 (18.3) | 40.2 (33.1) | 11.6 |
| $1 \times 50.0B$ | 17.4 (14.0) | 41.5 (33.6) | 9.6 |
| $2 \times 25.0B$ | 23.2 (19.5) | 42.1 (34.7) | 9.2 |
| $3 \times 16.7B$ | 23.8 (18.9) | 42.3 (34.4) | 11.2 |
| $4 \times 12.5B$ | **26.2 (21.3)** | 40.2 (32.5) | **12.8** |
| $5 \times 10.0B$ | 20.1 (17.7) | **43.9 (36.0)** | 10.4 |

tokens available. For example, $1 \times 50B$ indicates one repetition of the top 50B tokens, while $4 \times 12.5B$ denotes four repetitions of the top 12.5B tokens, ensuring that the selected tokens are of the best quality. Based on the results in the table, repeating the high-quality tokens four times ($4 \times 12.5B$) yields the best overall downstream performance across multiple evaluation metrics, showing the highest scores for HumanEval and EvoEval. Two repetitions ($2 \times 25.0B$) and three repetitions ($3 \times 16.7B$) also demonstrate strong performance, particularly in mbpp. Five repetitions ($5 \times 10.0B$) achieve the highest MBPP score but do not surpass the four repetitions in overall metrics. A single repetition ($1 \times 50.0B$) shows the least improvement compared to multiple repetitions.

## 4 RELATED WORK

### 4.1 CODE PRETRAINING CORPUS FOR LANGUAGE MODELS

Code data is essential to improving the reasoning capabilities of large language models (LLMs) (Aryabumi et al., 2024; Madaan et al., 2022; MA et al., 2024; Yang et al., 2024b; DeepSeek-AI et al., 2024b). Typically, researchers obtain massive code pretraining data by crawling from public platforms hosting code repositories such as GitHub (Li et al., 2023a; Rozière et al., 2024; Guo et al., 2024; Lozhkov et al., 2024; Mishra et al., 2024; DeepSeek-AI et al., 2024b). For example

The Stack v1 (Kocetkov et al., 2022) is a 3.1 TB dataset consisting of permissively licensed source code mined from GitHub in 30 programming languages. Its successor The Stack v2 (Lozhkov et al., 2024), built on the Software Heritage archive (Cosmo & Zacchiroli, 2017), is an order of magnitude larger, with a raw dataset of 67.5 TB spanning 619 programming languages. However, directly using these massive unfiltered code for pretraining is suboptimal, because the code documents may contain undesired contents or duplicates. Therefore, further preprocessing steps are needed to downscale the raw corpus, which can include deduplication (Li et al., 2023a; Rozière et al., 2024; Guo et al., 2024; Lozhkov et al., 2024; Mishra et al., 2024; DeepSeek-AI et al., 2024b; Team et al., 2024), PII (Personally Identifiable Information) redaction (Li et al., 2023a; Lozhkov et al., 2024; Mishra et al., 2024), benchmark decontamination (Li et al., 2023a; Lozhkov et al., 2024; Guo et al., 2024; DeepSeek-AI et al., 2024b), and model-based filtering (DeepSeek-AI et al., 2024b). As an example, StarCoder2 (Lozhkov et al., 2024) selects only 3 TB of data for pretraining from the 67.5 TB total data available in The Stack v2. The code pretraining corpus of Arctic-SnowCoder follows a similar preprocessing pipeline, comprising approximately 400B unique tokens from a mix of filtered The Stack v1 and GitHub crawls.

### 4.2 MODEL-BASED QUALITY FILTERING

In addition to common preprocessing steps like deduplication and heuristic filtering, a recent trend is using model-based quality filters to select high-quality pretraining data. Phi-1 (Gunasekar et al., 2023) employs a random forest classifier trained on top of the CodeGen (Nijkamp et al., 2023) embedding layer on GPT-4 annotations, to assess the educational value of files. This filter selects high-quality The Stack v1 and StackOverflow content, significantly enhancing coding performance. FineWeb-Edu (Penedo et al., 2024) employs a linear regressor built on `Snowflake-arctic-embed-m` (Merrick et al., 2024), an advanced embedding model based on BERT (Devlin et al., 2019). This regressor, trained on 400k Llama-3 (Dubey et al., 2024) annotations rating the educational value (0-5) of FineWeb dataset documents, significantly enhances STEM performance. DCLM-Baseline (Li et al., 2024) uses a `fastText` (Bojanowski et al., 2017) filter trained on positives from OpenHermes 2.5 (Teknium, 2023), high-scoring posts from `r/ExplainLikeImFive`, and random negatives. It outperforms FineWeb-Edu in top-10% selection. DeepSeek-Coder-V2 (DeepSeek-AI et al., 2024b) follows DeepSeek-Math (Shao et al., 2024) by leveraging a multi-stage `fastText`-based pipeline to recall high-quality code and math contents. Llama-3 (Dubey et al., 2024) uses `fastText` for recognizing text referenced by Wikipedia (Wikipedia contributors, 2004) and Roberta-based (Liu et al., 2019) classifiers trained on Llama-2 (Touvron et al., 2023) predictions. While prior work focuses on initial pretraining, Arctic-SnowCoder demonstrates that high-quality data from the pretraining corpus can significantly enhance model performance during continued pretraining. We are also the first to uncover the secret of data quality, revealing the importance of matching data distribution with downstream tasks.

### 4.3 HIGH-QUALITY CODE DATA FOR PRETRAINING

Phi-1 (Gunasekar et al., 2023) is one of the first to study the impact of high-quality code data. It first uses a random forest classifier to filter out high-quality code data from The Stack v1 and StackOverflow, and then creates synthetic textbook-like data and exercises using GPT-3.5 (OpenAI, 2022), showing significant coding performance with only 50B+ training tokens. DeepSeek-Coder-V2 (DeepSeek-AI et al., 2024b), pretrained for around 14T tokens in total, achieves state-of-the-art coding performance, with a multi-stage `fastText`-based (Bojanowski et al., 2017) pipeline to recall web-related code data as well as high-quality GitHub code. Arctic-SnowCoder utilizes a high-quality code annotator to extract high-quality code from pretraining datasets. It then generates synthetic files seeded from this high-quality data, adapting Magicoder OSS-Instruct (Wei et al., 2024b) into pretraining.

## 5 CONCLUSION

We introduce Arctic-SnowCoder-1.3B, a high-performing code model that underscores the critical importance of data quality in the pretraining process. Trained on 555B tokens, Arctic-SnowCoder-1.3B achieves competitive results with state-of-the-art small code models while using significantly fewer tokens. Our three-stage pretraining process begins with 500B tokens of general pretraining on

a raw code corpus, followed by 50B high-quality tokens scored by a quality annotator, and concludes with 5B tokens of synthetic data for further enhancement. This work demystifies the notion of high-quality data in code pretraining by demonstrating the key to high-quality data is its alignment with the distribution of downstream applications. Additionally, the paper offers practical guidelines for repo-level data grouping, learning rate scheduling, and the repetition of high-quality data, paving the way for more efficient and effective code model development.

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

## A EXAMPLES OF SYNTHETIC DATA

In this section, we present qualitative examples illustrating the disparities between the synthetic data generated during the enhanced pretraining stage and the high-quality data obtained through continued pretraining. The gray background code represents a high-quality code snippet annotated by the quality annotator, while the purple background code signifies the synthetic data of higher quality. A notable pattern of synthetic data is that it contains more documentation, natural language explanations, and comprehensive functionality. These features make the data more informative and boost the effectiveness of model learning.

### A.1 WAIT-UNTIL METHOD

```python
import time
from robot.utils import timestr_to_secs

class Wait:
    @staticmethod
    def until_true(condition, timeout, error_msg):
        """Helper to wait until given condition is met."""
        timeout = timestr_to_secs(timeout)
        max_wait = time.time() + timeout
        while True:
            if condition():
                break
            if time.time() > max_wait:
                raise AssertionError(error_msg)
            time.sleep(0.1)
```

```python
"""
Module: wait_until

This module provides a set of functions to wait until a given condition
    is met.
It is designed to help developers write more robust and reliable code by
    providing
a simple way to wait for certain conditions to be met before proceeding
    with the
execution of their code.

Author: [Your Name]
Date: [Today's Date]
"""

import time
from typing import Callable, Optional

def timestr_to_secs(timestr: str) -> float:
    """
    Convert a time string to seconds.

    Args:
    timestr (str): A string representing a time duration (e.g., "10s", "1
    m", "1h").

    Returns:
    float: The time duration in seconds.

    Raises:
    ValueError: If the input string is not in the correct format.

    Examples:
    >>> timestr_to_secs("10s")
```

```
      10.0
      >>> timestr_to_secs("1m")
      60.0
      >>> timestr_to_secs("1h")
      3600.0
      """
      if timestr.endswith("s"):
          return float(timestr[:-1])
      elif timestr.endswith("m"):
          return float(timestr[:-1]) * 60
      elif timestr.endswith("h"):
          return float(timestr[:-1]) * 3600
      else:
          raise ValueError("Invalid time string format")

def wait_until_true(
    condition: Callable[[], bool],
    timeout: str,
    error_msg: Optional[str] = None,
    poll_interval: float = 0.1,
) -> None:
      """
      Wait until the given condition is met.

      Args:
      condition (Callable[[], bool]): A function that returns a boolean
      value.
      timeout (str): A string representing the maximum time to wait (e.g.,
      "10s", "1m", "1h").
      error_msg (Optional[str]): An optional error message to raise if the
      condition is not met.
      poll_interval (float): The interval to wait between checks (default:
      0.1 seconds).

      Raises:
      AssertionError: If the condition is not met within the specified
      timeout.

      Examples:
      >>> def is_true():
      ...     return True
      >>> wait_until_true(is_true, "10s")
      >>> def is_false():
      ...     return False
      >>> try:
      ...     wait_until_true(is_false, "10s", "Condition not met")
      ... except AssertionError as e:
      ...     print(e)
      Condition not met
      """
      timeout_in_secs = timestr_to_secs(timeout)
      max_wait = time.time() + timeout_in_secs
      while True:
          if condition():
              break
          if time.time() > max_wait:
              if error_msg:
                  raise AssertionError(error_msg)
              else:
                  raise AssertionError("Condition not met")
          time.sleep(poll_interval)

def wait_until_false(
```

```
972          condition: Callable[[], bool],
973          timeout: str,
974          error_msg: Optional[str] = None,
975          poll_interval: float = 0.1,
976      ) -> None:
977          """
             Wait until the given condition is false.
978
979          Args:
980          condition (Callable[[], bool]): A function that returns a boolean
981          value.
982          timeout (str): A string representing the maximum time to wait (e.g.,
             "10s", "1m", "1h").
983          error_msg (Optional[str]): An optional error message to raise if the
984          condition is not met.
985          poll_interval (float): The interval to wait between checks (default:
986          0.1 seconds).
987
988          Raises:
             AssertionError: If the condition is not met within the specified
989          timeout.
990
991          Examples:
             >>> def is_false():
992          ...      return False
993          >>> wait_until_false(is_false, "10s")
994          >>> def is_true():
995          ...      return True
             >>> try:
996          ...      wait_until_false(is_true, "10s", "Condition not met")
997          ... except AssertionError as e:
998          ...      print(e)
999          Condition not met
1000         """
1001         wait_until_true(lambda: not condition(), timeout, error_msg,
1002         poll_interval)
1003
1004     if __name__ == "__main__":
1005         # Example usage:
1006         def is_true():
1007             return True
1008
             def is_false():
1009             return False
1010
1011         wait_until_true(is_true, "10s")
1012         try:
             wait_until_true(is_false, "10s", "Condition not met")
1013         except AssertionError as e:
1014             print(e)
1015
1016         wait_until_false(is_false, "10s")
1017         try:
             wait_until_false(is_true, "10s", "Condition not met")
1018         except AssertionError as e:
1019             print(e)
1020
1021
1022     A.2  QUICK SORT ALGORITHM
1023
1024     Array.prototype.quick_sort = function () {
1025         if (this.length < 2) { return this; }
```

```
1026      var pivot = this[Math.round(this.length / 2)];
1027
1028      return this.filter(x => x <  pivot)
1029              .quick_sort()
1030              .concat(this.filter(x => x == pivot))
1031              .concat(this.filter(x => x >  pivot).quick_sort());
1032  };
```

```
1033
1034  /**
1035   * Quick Sort Algorithm Implementation
1036   * ===============================
1037   *
1038   * This module provides a high-quality implementation of the quick sort
1039     algorithm.
1040   * It includes a stable and efficient sorting function, as well as a
1041     utility function
1042   * for generating random arrays for testing.
1043   */

1044  /**
1045   * Generates a random array of integers for testing.
1046   *
1047   * @param {number} length - The length of the array.
1048   * @param {number} min - The minimum value in the array.
1049   * @param {number} max - The maximum value in the array.
1050   * @returns {number[]} A random array of integers.
1051   */
1052  function generateRandomArray(length, min, max) {
1053    return Array.from({ length }, () => Math.floor(Math.random() * (max -
1054      min + 1)) + min);
1055  }

1056  /**
1057   * Swaps two elements in an array.
1058   *
1059   * @param {any[]} arr - The array.
1060   * @param {number} i - The index of the first element.
1061   * @param {number} j - The index of the second element.
1062   */
1063  function swap(arr, i, j) {
1064    [arr[i], arr[j]] = [arr[j], arr[i]];
1065  }

1066  /**
1067   * Partitions the array around a pivot element.
1068   *
1069   * @param {number[]} arr - The array.
1070   * @param {number} low - The low index.
1071   * @param {number} high - The high index.
1072   * @returns {number} The index of the pivot element.
1073   */
1074  function partition(arr, low, high) {
1075    const pivot = arr[high];
1076    let i = low - 1;
1077    for (let j = low; j < high; j++) {
1078      if (arr[j] < pivot) {
1079        i++;
         swap(arr, i, j);
       }
     }
     swap(arr, i + 1, high);
     return i + 1;
   }
```

```
1080    /**
1081     * Recursively sorts the array using the quick sort algorithm.
1082     *
1083     * @param {number[]} arr - The array.
1084     * @param {number} low - The low index.
1085     * @param {number} high - The high index.
1086     */
        function quickSort(arr, low, high) {
1087      if (low < high) {
1088        const pivotIndex = partition(arr, low, high);
1089        quickSort(arr, low, pivotIndex - 1);
1090        quickSort(arr, pivotIndex + 1, high);
1091      }
1092    }
1093
        /**
1094     * Sorts the array using the quick sort algorithm.
1095     *
1096     * @param {number[]} arr - The array.
1097     * @returns {number[]} The sorted array.
1098     */
        function sort(arr) {
1099      quickSort(arr, 0, arr.length - 1);
1100      return arr;
1101    }
```

## A.3 REVERSE WORDS

```
1105    # https://leetcode.com/problems/reverse-words-in-a-string-iii/
1106
1107    class Solution:
1108        def reverseWords(self, s: str) -> str:
1109            # return self.solution1(s)
1110            # return self.solution3(s)
1111            return self.solution2(s)
1112
        # Time O(n), Space O(n)
1113        def solution2(self, s):
1114            res = []
1115            word = []
1116
1117            for char in s:
1118                if char == ' ':
1119                    res.extend(reversed(word))
1120                    res.append(' ')
                    word = []
1121                else:
1122                    word.append(char)
1123
            res.extend(reversed(word))
1124
1125            return ''.join(res)
1126
1127        # Time O(n), Space O(n)
1128        def solution1(self, s):
1129            if len(s) <= 1:
1130                return s
1131
1132            counter = 0
            res = []
1133
            def extract_word(from_index, count):
```

```
1134                 for prev in range(count):
1135                     res.append(s[from_index - prev])
1136
1137         for idx, char in enumerate(s):
1138             if char == ' ':
1139                 extract_word(idx - 1, counter)
1140                 res.append(' ')
1141                 counter = 0
1142             else:
1143                 counter += 1
1144
1145         extract_word(len(s) - 1, counter)
1146
1147         return ''.join(res)
1148
1149     def solution3(self, s):
1150         return ' '.join([v[::-1] for v in s.split()])
```

```
# Reverse Words in a String III

class Solution:
    def reverse_words(self, s: str) -> str:
        """
        Reverses the order of characters in each word within a given
    string.

        Args:
            s (str): The input string.

        Returns:
            str: The string with each word's characters reversed.

        Example:
            >>> solution = Solution()
            >>> solution.reverse_words("Let's take LeetCode contest")
            "s'teL ekat edoCteeL tsetnoc"
        """
        return ' '.join(word[::-1] for word in s.split())

    def reverse_words_iterative(self, s: str) -> str:
        """
        Reverses the order of characters in each word within a given
    string using iteration.

        Args:
            s (str): The input string.

        Returns:
            str: The string with each word's characters reversed.

        Example:
            >>> solution = Solution()
            >>> solution.reverse_words_iterative("Let's take LeetCode
    contest")
            "s'teL ekat edoCteeL tsetnoc"
        """
        result = []
        word = []
        for char in s:
            if char == ' ':
                result.extend(reversed(word))
                result.append(' ')
                word = []
            else:
```

```
                   word.append(char)
        result.extend(reversed(word))
        return ''.join(result)

    def reverse_words_recursive(self, s: str) -> str:
        """
        Reverses the order of characters in each word within a given
    string using recursion.

        Args:
            s (str): The input string.

        Returns:
            str: The string with each word's characters reversed.

        Example:
            >>> solution = Solution()
            >>> solution.reverse_words_recursive("Let's take LeetCode
    contest")
            "s'teL ekat edoCteeL tsetnoc"
        """
        if not s:
            return s
        if s[0] == ' ':
            return ' ' + self.reverse_words_recursive(s[1:])
        if ' ' not in s:
            return s[::-1]
        space_index = s.index(' ')
        return s[:space_index][::-1] + ' ' + self.reverse_words_recursive
    (s[space_index + 1:])
```

## A.4 SKLEARN USAGE WITH JOBLIB

```
import joblib

from sklearn.feature_extraction.text import TfidfVectorizer
from sklearn.naive_bayes import MultinomialNB

class Model:
    def __init__(self, vocab_size=10_000):
        self.vocab_size = vocab_size
        self.clf = None
        self.vectorizer = None

    def train(self, X_train, y_train):
        self.vectorizer = TfidfVectorizer(max_features=self.vocab_size)
        X_train = self.vectorizer.fit_transform(X_train)

        self.clf = MultinomialNB()
        self.clf.fit(X_train, y_train)

    def predict(self, X):
        X = self.vectorizer.transform(X)

        self.y_pred = self.clf.predict(X)
        return self.y_pred

    def predict_proba(self, X):
        X = self.vectorizer.transform(X)

        self.y_proba = self.clf.predict_proba(X)
        return self.y_proba
```

```
1242
1243      def save_model(self):
1244          joblib.dump(self.vocab_size, "models/vocab.pkl")
1245          joblib.dump(self.vectorizer, "models/vecorizer.pkl")
1246          joblib.dump(self.clf, "models/model.pkl")
1247
1248      @staticmethod
1249      def load_model():
1250          model = Model()
1251          model.vocab_size = joblib.load("models/vocab.pkl")
1252          model.vectorizer = joblib.load("models/vecorizer.pkl")
1253          model.clf = joblib.load("models/model.pkl")
1254          return model
```

```
1255  import joblib
1256  from sklearn.feature_extraction.text import TfidfVectorizer
1257  from sklearn.naive_bayes import MultinomialNB
1258  from sklearn.model_selection import train_test_split
1259  from sklearn.metrics import accuracy_score, classification_report
1260
1261  class TextClassifier:
1262      def __init__(self, vocab_size=10_000):
1263          self.vocab_size = vocab_size
1264          self.vectorizer = TfidfVectorizer(max_features=vocab_size)
1265          self.clf = MultinomialNB()
1266
1267      def train(self, X, y):
1268          """Train the model on the given data.
1269
1270          Args:
1271              X (list): List of text samples.
1272              y (list): List of corresponding labels.
1273
1274          Returns:
1275              tuple: Training accuracy and classification report.
1276          """
1277          X_train, X_test, y_train, y_test = train_test_split(X, y,
1278      test_size=0.2, random_state=42)
1279          X_train = self.vectorizer.fit_transform(X_train)
1280          self.clf.fit(X_train, y_train)
1281
1282          X_test = self.vectorizer.transform(X_test)
1283          y_pred = self.clf.predict(X_test)
1284          accuracy = accuracy_score(y_test, y_pred)
1285          report = classification_report(y_test, y_pred)
1286          return accuracy, report
1287
1288      def predict(self, X):
1289          """Make predictions on the given data.
1290
1291          Args:
1292              X (list): List of text samples.
1293
1294          Returns:
1295              list: List of predicted labels.
          """
          X = self.vectorizer.transform(X)
          return self.clf.predict(X)

      def save_model(self, path="models/"):
          joblib.dump(self.vectorizer, path + "vectorizer.pkl")
          joblib.dump(self.clf, path + "model.pkl")

      @staticmethod
```

```
1296    def load_model(path="models/"):
1297        model = TextClassifier()
1298        model.vectorizer = joblib.load(path + "vectorizer.pkl")
1299        model.clf = joblib.load(path + "model.pkl")
1300        return model
```

## A.5  JAVA TEXT VIEW BINDING

```
package android.databinding.testapp.vo;

import android.databinding.Bindable;

public class AutoCompleteTextViewBindingObject extends
    BindingAdapterBindingObject {
    @Bindable
    private int mPopupBackground;

    @Bindable
    private int mCompletionThreshold = 1;

    public int getCompletionThreshold() {
        return mCompletionThreshold;
    }

    public int getPopupBackground() {
        return mPopupBackground;
    }

    public void changeValues() {
        mPopupBackground = 0xFF23456;
        mCompletionThreshold = 5;
        notifyChange();
    }
}
```

```
package com.example.databinding;

import android.databinding.Bindable;
import android.databinding.Observable;

/**
 * Represents a binding object for AutoCompleteTextView.
 */
public class AutoCompleteTextViewBindingObject extends Observable {

    // Default values
    private static final int DEFAULT_POPUP_BACKGROUND = 0xFFFFFFFF;
    private static final int DEFAULT_COMPLETION_THRESHOLD = 1;

    @Bindable
    private int mPopupBackground;

    @Bindable
    private int mCompletionThreshold;

    /**
     * Constructs a new AutoCompleteTextViewBindingObject with default
     values.
     */
    public AutoCompleteTextViewBindingObject() {
        this(DEFAULT_POPUP_BACKGROUND, DEFAULT_COMPLETION_THRESHOLD);
    }
```

```java
    /**
     * Constructs a new AutoCompleteTextViewBindingObject with the
    specified values.
     *
     * @param popupBackground The background color of the popup.
     * @param completionThreshold The minimum number of characters to
    show the popup.
     */
    public AutoCompleteTextViewBindingObject(int popupBackground, int
    completionThreshold) {
        this.mPopupBackground = popupBackground;
        this.mCompletionThreshold = completionThreshold;
    }

    /**
     * Gets the background color of the popup.
     *
     * @return The background color of the popup.
     */
    public int getPopupBackground() {
        return mPopupBackground;
    }

    /**
     * Sets the background color of the popup.
     *
     * @param popupBackground The new background color of the popup.
     */
    public void setPopupBackground(int popupBackground) {
        if (this.mPopupBackground != popupBackground) {
            this.mPopupBackground = popupBackground;
            notifyPropertyChanged(BR.popupBackground);
        }
    }

    /**
     * Gets the minimum number of characters to show the popup.
     *
     * @return The minimum number of characters to show the popup.
     */
    public int getCompletionThreshold() {
        return mCompletionThreshold;
    }

    /**
     * Sets the minimum number of characters to show the popup.
     *
     * @param completionThreshold The new minimum number of characters to
     show the popup.
     */
    public void setCompletionThreshold(int completionThreshold) {
        if (this.mCompletionThreshold != completionThreshold) {
            this.mCompletionThreshold = completionThreshold;
            notifyPropertyChanged(BR.completionThreshold);
        }
    }

    /**
     * Changes the values of the binding object.
     *
     * @param popupBackground The new background color of the popup.
     * @param completionThreshold The new minimum number of characters to
     show the popup.
     */
```

```
1404    public void changeValues(int popupBackground, int completionThreshold
1405    ) {
1406        setPopupBackground(popupBackground);
1407        setCompletionThreshold(completionThreshold);
1408    }
1409 }
```

