# OpenReview forum: "Arctic-SnowCoder: Demystifying High-Quality Data in Code Pretraining"
_ICLR.cc/2025/Conference — Submitted to ICLR 2025_

### Official Review · Reviewer_YFrR · 2024-10-28

**Soundness:** 3
**Presentation:** 3
**Contribution:** 2
**Rating:** 6
**Confidence:** 4

**Summary:**

The authors provide practical findings through pretraining a small code language model (Code LM), which achieves SOTA performance on representative coding benchmarks like HumanEval, MBPP, and BigCodeBench, among the Code LMs with similar sizes.

**Strengths:**

The authors documented various valuable practices of pretraining Code LMs from scratch, which can inspire future work in this direction:
- Training a BERT-based classifier to annotate code quality, which is very efficient compared to any LLMs-as-Judges approaches.
- Using re-warmup as the learning rate schedule is quite novel.
-  While previous studies [1] suggest that deduplicating the data will result in better model performance, training on the repeated high-quality code data can further improve the coding capabilities.

The authors also shared a few interesting findings:
- "Textbook" is not all you need, so improving "educational value" in the training data may not be optimal.
- Re-warmup performs much better than other conventional schedules, such as linear and constant.
- When the number of high-quality tokens is limited to 50B, the setup of 12.5B with four repetitions could be more optimal.

[1] Lee, K., Ippolito, D., Nystrom, A., Zhang, C., Eck, D., Callison-Burch, C., & Carlini, N. (2022, May). Deduplicating Training Data Makes Language Models Better. In Proceedings of the 60th Annual Meeting of the Association for Computational Linguistics (Volume 1: Long Papers) (pp. 8424-8445).

**Weaknesses:**

There are a few weaknesses in this paper:
- The current evaluations are Python-only, and the evaluation on multilingual (programming languages) code generation may share more interesting findings.
- The authors only study the 1.3B models, which is considered a bit small. While I understand that pretraining LMs is very costly, is it possible for the authors to provide more motivation for studying the ~1B models?
- It is most likely that the data cannot be opened due to legal constraints.

**Questions:**

1. Regarding the openness of this work, will the authors consider making the data pipeline and models publicly available? This will greatly help future studies on Code LM pretraining.
2. Regarding the evaluation, can the authors provide some explanations as to why BigCodeBench results are omitted in most of the tables?
3. Can the authors share more insights on how current setups documented in the paper can be generalized to larger Code LMs (e.g., 10B+)?

---

> ### Author Response · Authors · 2024-11-23
>
> Thank you for your detailed review! We provide our responses as follows.
>
> > Q1: Regarding the openness of this work, will the authors consider making the data pipeline and models publicly available? This will greatly help future studies on Code LM pretraining.
>
> Great point. We will try the best to open source the pipeline, model, and the annotator to facilitate future research in code LLM pretraining. However, due to proprietary reasons, this process would be long and we hope the reviewers can understand. We have also revised our paper to cover more technical details and insights.
>
> > Q2: Regarding the evaluation, can the authors provide some explanations as to why BigCodeBench results are omitted in most of the tables?
>
> The primary reason is that BigCodeBench is a very comprehensive benchmark covering 1,140 task instances, each of which can involve multiple function calls from external libraries, and this makes evaluation on every checkpoint costly with our resource constraint. We are happy to incorporate more evaluation on BigCodeBench in the final revision.
>
> > Q3: Can the authors share more insights on how current setups documented in the paper can be generalized to larger Code LMs (e.g., 10B+)?
>
> Good question. While we are unable to verify the exact effectiveness of 10B+ code models due to the cost of pretraining them, we believe the data insights from our work can generalize to larger code models as well. Specifically, for larger code LMs, we can make the annotator cover more scenarios since they are able to hold more knowledge internally. For bigger models, it can also be beneficial to train for a longer token horizon.

---

> > ### Author Response · Authors · 2024-11-23
> >
> > > Q4: The current evaluations are Python-only...
> >
> > Thanks for your suggestion! We will include more multi-lingual evaluations in the final revision.
> >
> > > Q5: ...more motivation for studying the ~1B models?
> >
> > Great point. The primary reason is that we don't have enough resources to scale this result further to a larger model. Meanwhile, small code LMs with ~1B parameters can serve as a good proxy for the data quality and make experimenting with various different configurations affordable.
> >
> > > Q6: It is most likely that the data cannot be opened due to legal constraints.
> >
> > While there are indeed legal constraints, as explained in Q1, we will make every effort to open source our research artifacts to benefit the research community.

---

> > > ### Comment · Reviewer_YFrR · 2024-11-24
> > >
> > > Thank you for the response! I remain positive about this paper.

---

### Official Review · Reviewer_Nj9G · 2024-10-29

**Soundness:** 2
**Presentation:** 2
**Contribution:** 3
**Rating:** 5
**Confidence:** 3

**Summary:**

This paper introduces a  method that includes three different pre-training phases, combined with iterative improvements to the quality of training data. It presents a code model—Arctic-SnowCoder-1.3B—which demonstrates competitive performance compared to current small code models, while significantly reducing the number of tokens used. The paper also provides guidelines for repo-level data grouping, learning rate scheduling, and emphasizes the importance of high-quality data.

**Strengths:**

1. This paper proposes a method for improving the performance of pre-training models by focusing on multi-stage data quality enhancement. It introduces a high-performing code model with low token usage.
2. Additionally, the paper analyzestraining strategies, including emphasizing the preparation of training data files and the characteristics of learning rate scheduling.

**Weaknesses:**

This paper primarily focuses on techniques for enhancing and filtering the quality of code training data, with a key emphasis on how high-quality, filtered data improves model performance. However, an important question arises: could this improvement come at a cost, such as reduced generalization ability on non-target domain tasks?

Additionally, the paper should review some existing techniques for improving training data quality and, where appropriate, include comparative analyses to demonstrate the advantages of the proposed method.

**Questions:**

There are some questions after reviewing the paper:
1. In line 190,  "increase the Python mix ratio to approximately 50% while keeping the proportions of the other languages unchanged.", why is Python set as the primary language data, and how can it be adjusted for other languages?
2. In line 295,  "We can observe that the second approach, which we finally adopt in general pretraining, performs significantly better than the first one.", could you further explain the reason behind this conclusion?
3. In line 351, "the key to high-quality data is essentially the alignment with downstream application distributions.", what is the difference between alignment and fine-tuning of pre-trained models? And what are the advantages of the method proposed compared to fine-tuning techniques?

---

> ### Author Response · Authors · 2024-11-23
>
> Thank you for your valuable review and suggestions! We provide our response as follows.
>
> > Q1: ...why is Python set as the primary language data, and how can it be adjusted for other languages?
>
> Thank you for the great point. The primary reason we set Python as a focus during enhanced pretraining is to align with prior work, such as StarCoder and CodeLlama, which emphasize specific Python variants. While those approaches dedicate a final fine-tuning stage exclusively to Python, our method balances this by adjusting the Python mix ratio to 50%, while preserving the relative proportions of other languages. This ensures that Python is adequately emphasized without overshadowing other languages. It’s important to note that in the earlier phases of pretraining, we adhered to a natural distribution of languages without prioritizing Python.
>
> > Q2: ...performs significantly better than the first one.", could you further explain the reason behind this conclusion?
>
> The primary reason for improved performance when grouping by language before repository is that grouping by repositories can lead to training instances containing mixed file types, such as configuration files and programming files. During training, we align the compute, meaning that the "grouping by repositories" approach processes fewer tokens specifically from programming files. Moreover, because files are randomly ordered, code files from different languages are often unrelated. Consequently, each training example may end up including two completely unrelated files, which can negatively impact learning.
>
> A promising hybrid approach could involve grouping files by language within each repository. This method ensures that training examples can include multiple programming language files while maintaining the cohesion of files in the same language within each group.
>
> We have put these explanations in the revised paper.
>
> > Q3: ...what is the difference between alignment and fine-tuning of pre-trained models?...
>
> We would like to clarify that the term "alignment" in this context specifically refers to "distribution matching" and is distinct from the use of the term "alignment" in the context of training paradigms. We have revised the wording in our manuscript accordingly for better clarity.

---

### Official Review · Reviewer_fT8z · 2024-11-04

**Soundness:** 3
**Presentation:** 3
**Contribution:** 3
**Rating:** 6
**Confidence:** 3

**Summary:**

The authors emphasize the critical role of high-quality data in the effective pretraining of language models, particularly within the code domain, while noting that the precise definition of "high-quality" remains inadequately explored. To address this, they introduce Arctic-SnowCoder-1.3B, a data-efficient base code model pretrained on 555 billion tokens through three phases of progressively refined data. The first phase involves general pretraining with 500 billion standard-quality code tokens, processed through basic filtering, deduplication, and decontamination. The second phase continues with 50 billion high-quality tokens, selected from the first phase by a BERT-style quality annotator trained to distinguish good code from random data, using high-quality code files and instruction data from Magicoder and StarCoder2-Instruct. The final phase employs 5 billion synthetic tokens generated by Llama-3.1-70B, using phase two data as seeds and adapting the Magicoder approach for pretraining. Despite the limited dataset, Arctic-SnowCoder achieves state-of-the-art performance on BigCodeBench, outperforming similarly sized models trained on up to 1 trillion tokens, including a 36% improvement over Phi-1.5-1.3B. Across various benchmarks, Arctic-SnowCoder-1.3B performs better than StarCoderBase-3B pretrained on 1 trillion tokens and matches the performance of leading small base code models trained on trillions of tokens.

**Strengths:**

+ Important Area.

The authors address a critical aspect of language model development—high-quality data in the code domain—which is essential for improving model performance and applicability.



+ Good Performance on BigCodeBench

Arctic-SnowCoder-1.3B demonstrates good results, achieving state-of-the-art performance on BigCodeBench and surpassing similarly sized models trained on up to 1 trillion tokens, including a notable 36% improvement over Phi-1.5-1.3B.

**Weaknesses:**

1. Limited Novelty: While the use of a data annotator to extract high-quality data for pretraining is a valuable approach, it is not entirely novel. Similar methodologies have been employed, such as using GPT-4 as a data annotator. This raises questions about the uniqueness of the authors' contributions.

2. Missing Baselines: The evaluation would benefit from the inclusion of additional baselines, such as OpenAI's GPT models. Comparing or discussing these established models would provide a more comprehensive context for assessing Arctic-SnowCoder's performance and highlight its relative strengths and weaknesses.

3. Lower than Phi-1.5-1.3B on HumanEval+ MBPP+ and EvoEval: Despite achieving strong performance on BigCodeBench, Arctic-SnowCoder-1.3B underperforms compared to Phi-1.5-1.3B on more general code generation tasks, such as HumanEval+, MBPP+, and EvoEval. Interestingly, Phi-1.5-1.3B achieved better results with less training data, which suggests that Arctic-SnowCoder's specialized pretraining on high-quality code tokens may not necessarily translate into better generalization across a broader range of benchmarks.

**Questions:**

The authors address a critical aspect of language model development—high-quality data in the code domain—which is essential for improving model performance and applicability. For the experiments, Arctic-SnowCoder-1.3B demonstrates good results, achieving state-of-the-art performance on BigCodeBench and surpassing similarly sized models trained on up to 1 trillion tokens, including a notable 36% improvement over Phi-1.5-1.3B.

However, I have three concerns:

1. Limited Novelty: While the use of a data annotator to extract high-quality data for pretraining is a valuable approach, it is not entirely novel. Similar methodologies have been employed, such as using GPT-4 as a data annotator. This raises questions about the uniqueness of the authors' contributions.

2. Missing Baselines: The evaluation would benefit from the inclusion of additional baselines, such as OpenAI's GPT models. Comparing or discussing these established models would provide a more comprehensive context for assessing Arctic-SnowCoder's performance and highlight its relative strengths and weaknesses.

3. Lower than Phi-1.5-1.3B on HumanEval+ MBPP+ and EvoEval: Despite achieving strong performance on BigCodeBench, Arctic-SnowCoder-1.3B underperforms compared to Phi-1.5-1.3B on more general code generation tasks, such as HumanEval+, MBPP+, and EvoEval. Interestingly, Phi-1.5-1.3B achieved better results with less training data, which suggests that Arctic-SnowCoder's specialized pretraining on high-quality code tokens may not necessarily translate into better generalization across a broader range of benchmarks.

---

> ### Author Response · Authors · 2024-11-23
>
> Thank you for your review and suggestions! We address your questions as follows.
>
> > Q1: ...Similar methodologies have been employed, such as using GPT-4 as a data annotator. This raises questions about the uniqueness of the authors' contributions.
>
> We agree that there are many approaches relying on GPT-4 for generating instruction tuning data. However, using GPT-4 for annotating the huge pretraining corpora is cost-prohibitive and does not scale effectively. Instead, our method leverages a BERT-based annotator, which is significantly more efficient for both filtering and annotation purposes.
>
> The key innovation lies in combining data filtering and data synthesis—a novel approach that we first proposed. Additionally, we provide actionable insights, such as optimal re-warmup strategies during continued pretraining, which align with Reviewer 4fJs’s feedback.
>
> > Q2: ...The evaluation would benefit from the inclusion of additional baselines, such as OpenAI's GPT models...
>
> Thanks for your suggestion. We did not include GPT models at the first place primarily because SnowCoder focuses on small code models that are open-source. Given the close nature of OpenAI models, it is difficult to draw conclusions from comparisons with them. The primary goal of SnowCoder is to provide insights and offer practical guidelines for researchers working on pretraining, and to benefit future pretraining work. Here are the additional evaluation results for OpenAI's GPT series and Anthropic's Claude series:
>
> |Model|HumanEval+|MBPP+|BigCodeBench|
> |-|-|-|-|
> |GPT-4o|87.2|72.2|56.1|
> |GPT-4-Turbo|81.7|73.3|53.2|
> |GPT-4o-mini|83.5|72.2|51.8|
> |GPT-3.5-Turbo|70.7|69.7|44.9|
> |Claude-3.5-Sonnet|81.7|74.3|52.7|
> |Claude-3-Opus|77.4|73.3|51.5|
> |Claude-3-haiku|68.9|68.8|44.8|
>
> We are happy to include them in the final revision if the reviewer believes they are important.
>
> > Q3: Lower than Phi-1.5-1.3B on HumanEval+ MBPP+...
>
> Great question! It’s important to note that HumanEval+ comprises just 164 self-contained Python problems, making it a relatively narrow benchmark for evaluating general-purpose code generation. In contrast, BigCodeBench is far more comprehensive and practical. It challenges LLMs to tackle 1,140 fine-grained tasks across 139 libraries and 7 domains, requiring the ability to invoke multiple function calls as tools.
>
> Moreover, Phi-1.5-1.3B uses GPT-3.5 and GPT-4 to generate over 30 billion textbook-like tokens —this is a significantly more expensive approach than SnowCoder. At current pricing for GPT4-o models at `$2.5` per 1M tokens, this will take `$75,000` to generate one time; usually when experimenting, one can easily expect to pay 2x that or more. This is a significant expense. This dataset is heavily skewed toward HumanEval-like problems, which may explain its advantage on such benchmarks. In comparison, Arctic-SnowCoder-1.3B’s specialized pretraining on high-quality code tokens prioritizes real-world applicability over narrowly optimized generalization to benchmarks like HumanEval+.

---

> > ### Comment · Reviewer_fT8z · 2024-11-27
> > **Good**
> >
> > The answers address my concerns well. I changed my score to 6.

---

### Official Review · Reviewer_1VWJ · 2024-11-04

**Soundness:** 3
**Presentation:** 3
**Contribution:** 2
**Rating:** 5
**Confidence:** 5

**Summary:**

This paper introduces Arctic-SnowCoder-1.3B, a small code model trained on 555B tokens through a meticulously designed three-phase pretraining process. The first phase involves general pretraining with 500B tokens of standard-quality code, filtered and deduplicated. The second phase refines this with 50B tokens of high-quality code, identified using a BERT-based quality annotator trained on positive examples from curated open-source repositories and instruction datasets. In the final phase, 5B tokens of synthetic data are generated using Llama-3.1-70B, seeded from the high-quality data to further enhance model performance. The model achieves state-of-the-art results on BigCodeBench, significantly outperforming larger models on practical and challenging programming benchmarks such as HumanEval+ and MBPP+. The paper highlights the importance of progressively improving data quality and aligning it with downstream tasks, offering comprehensive evaluations, ablation studies, and practical insights into optimal pretraining strategies, such as learning rate schedules and data repetition, to maximize the efficiency of smaller language models in code generation tasks.

**Strengths:**

Arctic-SnowCoder demonstrates remarkable strengths among small size model, particularly in achieving state-of-the-art results on BigCodeBench with a 36% performance improvement over Phi-1.5-1.3B, despite using only 555B tokens compared to models trained on trillions of tokens. Arctic-SnowCoder-1.3B outperforms StarCoderBase-3B across all benchmarks and surpasses StarCoder2-3B, trained on over 3.3T tokens, on HumanEval+ with a score of 28.0 compared to 27.4. The model also achieves competitive results on MBPP+ (42.9) and EvoEval (18.0), showing that it can match or exceed the performance of larger models like StableCode-3B and Granite-Code-Base-3B, which are trained on 1.3T and 4.5T tokens, respectively. These results, combined with thorough ablation studies, highlight the effectiveness of its three-phase pretraining strategy, focusing on high-quality and synthetic data, while providing concrete evidence of its efficiency and superior performance in practical and complex coding tasks.

**Weaknesses:**

While the synthetic data significantly boosts performance, as seen in the 36% improvement over Phi-1.5-1.3B on BigCodeBench, an overreliance on synthetic data risks skewing the model’s understanding of practical coding tasks. Additionally, the performance on HumanEval+ (28.0) and MBPP+ (42.9), although impressive, shows only incremental improvements over models like StarCoder2-3B (27.4 on HumanEval+ and 49.2 on MBPP+), indicating room for optimization in handling more complex or diverse programming tasks. The quality annotator, trained on specific curated datasets, could introduce biases that may not adequately represent broader coding practices, potentially limiting its effectiveness across all programming domains.

The paper’s approach to handling repo-level data in the general pretraining phase is insightful but has room for further exploration. The authors compare two methods: grouping files by repository names and by language before repository. They conclude that partitioning by language yields better results, as evidenced by improved scores on HumanEval+ (17.1 vs. 12.8), MBPP+ (33.9 vs. 30.7), and EvoEval (7.4 vs. 7.0). This method ensures that training documents are more focused and homogenous, which likely aids the model in learning language-specific patterns effectively. However, this method might overlook the potential benefits of cross-language learning, especially in multi-language projects where inter-language interactions are critical. Future work could explore hybrid approaches that maintain language-specific grouping but occasionally incorporate multi-language contexts to enhance the model’s ability to handle real-world, polyglot codebases. Additionally, more granular investigations into the impact of repository size and the diversity of code snippets within a repository could provide deeper insights into optimizing repo-level data grouping for enhanced model performance.

In addition, my concern is that the paper presents compelling results among small language models, particularly with its strong performance on benchmarks like BigCodeBench and HumanEval+. However, the underlying reasons for achieving such high performance despite the relatively small training dataset (555B tokens) are not fully unpacked. While the authors attribute the success to the progressive refinement of data quality and the use of synthetic data, the detailed mechanisms by which these factors translate into superior model performance remain somewhat opaque.

Finally, I suggest evaluating the model on the CodeMMLU benchmark, which could provide a broader assessment of the model’s capabilities across a diverse set of coding tasks, thereby offering more comprehensive insights into its strengths and potential areas for improvement.

[1] CodeMMLU: A Multi-Task Benchmark for Assessing Code Understanding Capabilities of CodeLLMs, https://arxiv.org/abs/2410.01999

**Questions:**

1) Could you provide more detailed analysis or ablation studies on how the quality annotator specifically improves the model’s learning? What are the key features or patterns it identifies that contribute most to the performance boost?

2) How does the synthetic data generated by Llama-3.1-70B differ in quality or characteristics from the high-quality tokens selected by the annotator? Could you provide examples or metrics that highlight these differences?

3) Your results suggest that grouping by language before repository improves performance. Could you elaborate on why this approach works better? Have you considered any hybrid methods that combine cross-language learning with language-specific training?

4) Given the success of Arctic-SnowCoder-1.3B with 555B tokens, how do you envision scaling this approach for larger models or different domains? Are there diminishing returns or unique challenges you anticipate?

5) The paper focuses on benchmarks like BigCodeBench and HumanEval+. How do you ensure these benchmarks reflect real-world programming challenges? Have you considered any additional metrics or benchmarks that might better capture practical coding scenarios?

---

> ### Author Response · Authors · 2024-11-23
>
> Thank you for your valuable review and suggestions! We put our responses to your questions as follows.
>
> > Q1: ...how the quality annotator specifically improves the model’s learning? What are the key features or patterns it identifies that contribute most to the performance boost?
>
> Good questions. Please kindly refer to Table 4, where we studied different data combinations used to train the quality annotator. In Figure 2, we showed that the downstream performance on a given benchmark directly correlates with the ROC-AUC score of the quality annotator on that benchmark. A higher ROC-AUC score indicates that the annotator can rank the corresponding downstream distribution higher. And this is how we conclude that the key to high-quality data is its alignment with the distribution of downstream applications.
>
> > Q2: How does the synthetic data generated by Llama-3.1-70B differ in quality or characteristics from the high-quality tokens...
>
> Great point. We leveraged Llama-3.1-70B to rephrase a lower-quality code file into a higher-quality variant, where one of the key instructions is to let the model make sure the code file can improve a developer's ability in terms of language comprehension, reasoning, algorithms, and mathematics. Due to the text length limit in rebuttal, we have put several concrete synthetic data examples in Appendix A in our revised paper. From the examples, we can observe that a notable pattern of synthetic data is that it contains more documentation, natural language explanations, and comprehensive functionality. These features make the data more informative and boost the effectiveness of model learning.
>
> > Q3: Your results suggest that grouping by language before repository improves performance. Could you elaborate on why this approach works better?...
>
> The primary reason for improved performance when grouping by language before repository is that grouping by repositories can lead to training instances containing mixed file types, such as configuration files and programming files. During training, we align the compute, meaning that the "grouping by repositories" approach processes fewer tokens specifically from programming files. Moreover, because files are randomly ordered, code files from different languages are often unrelated. Consequently, each training example may end up including two completely unrelated files, which can negatively impact learning.
>
> A promising hybrid approach could involve grouping files by language within each repository. This method ensures that training examples can include multiple programming language files while maintaining the cohesion of files in the same language within each group.
>
> We have put these explanations in our revised paper.
>
> > Q4: ...how do you envision scaling this approach for larger models or different domains...?
>
> We believe that further scaling up the training tokens can lead to significant performance improvements, as evidenced by the trends observed during the continued pretraining and enhanced pretraining phases. However, our experiments indicate the importance of aligning the training distribution with downstream distributions. Misalignment in data distribution poses a critical challenge during the pretraining of SnowCoder, as it often results in diminishing returns. Specifically, while the validation loss continues to decrease, downstream performance plateaus or fails to improve.
>
> > Q5: ..How do you ensure these benchmarks reflect real-world programming challenges...
>
> Great point. We choose HumanEval+ due to its wide adoption and its robustness compared to HumanEval. While HumanEval+ primarily focuses on self-contained coding tasks with just 164 problems, BigCodeBench is a comprehensive coding benchmark that challenges LLMs to invoke multiple function calls as tools from 139 libraries and 7 domains for 1,140 fine-grained tasks. Also, the selected benchmarks are well suited for evaluating pretraining models, while most other real-world programming challenges will require instruction tuning, which is not the primary focus of this paper. We believe the combination of the two benchmarks can cover most practical coding scenarios.

---

> > ### Author Response · Authors · 2024-11-23
> >
> > > Q6:  …an overreliance on synthetic data risks skewing the model’s understanding of practical coding tasks...
> >
> > While Arctic-SnowCoder shows incremental downstream performance over models like StarCoder2-3B, we would like to highlight that Arctic-SnowCoder achieves such performance with much less training compute (555B training tokens), compared to StarCoder2-3B which is trained for 3.3-4.3T tokens. This indicates that our data is of higher quality. At the meantime, to ensure the diversity of programming tasks, we used BigCodeBench, a comprehensive coding benchmark that challenges LLM to invoke multiple function calls as tools from 139 libraries and 7 domains for 1,140 fine-grained tasks.
> >
> > > Q7: The paper’s approach to handling repo-level data in the general pretraining phase is insightful but has room for further exploration…
> >
> > Great point. Please kindly refer to Q2 where we explained one possible hybrid approach. We have included this point in our revised manuscript.
> >
> > > Q8: …the underlying reasons for achieving such high performance despite the relatively small training dataset (555B tokens) are not fully unpacked…
> >
> > As we explained in Q1, the main secret of training a good data quality annotator is its ability to distinguish code that matches downstream distributions. Regarding the synthetic data, please kindly refer to Q2, where we explained its difference and why it is effective.

---

### Author Response · Authors · 2024-11-23

We deeply appreciate all the reviewers for their insightful feedback and suggestions for our work. In our responses below, we address each primary question (denoted as Q) raised by the individual reviewers.

We have revised our paper to incorporate the important comments raised by the reviewers. Given the significant cost of pretraining and our resource constraints, we are unable to conduct additional training experiments during the rebuttal period. However, we are committed to addressing all other aspects of the feedback to the best of our ability. Additionally, we will conduct more experiments after the rebuttal period and put them in the final revision. Should there be any misunderstandings of the questions, please kindly let us know; we are eager to communicate with all the reviewers throughout the discussion period.

---

### Meta-Review · Area_Chair_aYvT · 2024-12-16

**Metareview:**

This work presents a 1.3B coding-specialized model that demonstrates strong performance on benchmarks such as HumanEval, MBPP, and others. A key contribution lies in the innovative data curation process, leveraging a lightweight BERT classifier to filter high-quality data and later generating synthetic data using LLama-3.1-70B.

While reviewers recognize the value of the provided artifacts, including the model and data pipeline for the open-source community, the main concerns center on whether all components will be fully released due to proprietary constraints. Despite the authors expressing their intention to make a best-effort attempt to share as much as possible, the lack of a guarantee on the release has led to the leaning towards the rejection.

**Additional Comments On Reviewer Discussion:**

NA

---

### Decision · Program_Chairs · 2025-01-22

Reject